# OpenReview forum: "Baseline Defenses for Adversarial Attacks Against Aligned Language Models"
_ICLR.cc/2024/Conference — Submitted to ICLR 2024_

### Official Review · Reviewer_UELn · 2023-10-30

**Soundness:** 3 good
**Presentation:** 3 good
**Contribution:** 2 fair
**Rating:** 5
**Confidence:** 4

**Summary:**

This paper presents a study of the vulnerabilities associated with large language models (LLMs), specifically focusing on jailbreaking attacks that aim to bypass alignment and moderation mechanisms. The work mainly adopts an empirical approach, evaluating several baseline defenses under different threat models and settings, and shows the difference between the attacks/defenses in the domain of LLMs and that in conventional domains (e.g., computer vision).

**Strengths:**

- The paper considers three categories of defenses: detection via perplexity filtering, input preprocessing (including paraphrasing and re-tokenization), and adversarial training.
- Using the computational cost (rather than the perturbation magnitude) as a constraint for the adversary seems interesting.
- It discusses the robustness-performance trade-off in each defense strategy, providing insights into their feasibility and effectiveness.

**Weaknesses:**

- The main findings of the paper seem to be the difference between the attacks/defenses in LLMs and other domains (e.g., computer vision). This is not news. There are many studies in the literature that have pointed out such difference.
- The evaluation mainly focuses on a single attack (Zou et al. 2023). It is unclear whether the findings are biased by this setting. It is suggested to consider more diverse jailbreaking attacks.
- While the paper sets a good foundation for understanding baseline defenses, it could benefit from a clearer roadmap or suggestions for future research directions.
- The presentation could also be improved. It is suggested to summarize the findings in each part of the evaluation, which right now are scattered around.

**Questions:**

- How do the findings generalize to attacks other than Zou et al. 2023?

---

> ### Author Response · Authors · 2023-11-16
> **Response to Reviewer UELn**
>
> Thank you for your time, UELn!
>
> > future directions
>
> In the last paraphrase of the conclusion, we list the future directions of the paper. We highlight the following questions
>
> "...leaves us with a number of big open questions. (i) What defenses can be reliably deployed, with only minimal impact on benign performance? (ii) Do adaptive attacks that bypass these defenses transfer from surrogate to target models in the gray-box setting? (iii) Can we find good approximations to robust optimization objectives that allow for successful adversarial training? (iv) Can we theoretically bound, or certify, the minimal computational budget required for an attack against a given (gray-box) defense, thereby guaranteeing a level of safety based on computational complexity? Most importantly, (v) can discrete text optimizers be developed that significantly improve the effectiveness of adversarial attacks? If so, this would return LLM security to a state more like that of computer vision"
>
>
> > mainly focuses on a single attack
>
> We have added a non-universal threat model, where the attacker has access to the model weights and optimizes an attack for one string. However, the attack does not have access to the system, so the defender can put preprocessing and post processing defenses in the pipeline. With this threat model, we optimize one string on the attacked model (Vicuna-v1.1) exploring the adaptive attack.
>
> | PPL Weight | Non-Attack ASR | ASR w/o filter | Passed PPL | Passed WinPPL | Total ASR (PPL Filter) | Total ASR (WinPPL Filter) |
> | ---------- | -------------- | -------------- | ---------- | ------------- | ---------------------- | ------------------------- |
> | 0          | 5%             | 100%           | 0%         | 0%            | 0%                     | 0%                        |
> | 0.1        | 5%             | 56%            | 50%        | 12%           | 14%                    | 0%                        |
> | 0.2        | 5%             | 12%            | 94%        | 56%           | 12%                    | 2%                        |
>
>
> > literature
>
> We would be glad to include these works in our related work discussion if you could to point us to these papers. We already provided some overview of the literature for attacks against encoder models. However, we believe Zou et al. building off of Wallace et al. showed the true strength of these attacks on large-scale decoder models for text generation.
>
> Please let us know if there are any follow-up questions.

---

> > ### Comment · Reviewer_UELn · 2023-11-23
> >
> > I appreciate the authors' response and added experiments, which help clarify some of my queries. However, I still feel it's risky to draw conclusions about the whole defense landscape based on one (or two) attacks. In light of this, I am inclined to maintain my original rating.

---

### Official Review · Reviewer_Ywox · 2023-10-31

**Soundness:** 4 excellent
**Presentation:** 4 excellent
**Contribution:** 3 good
**Rating:** 8
**Confidence:** 3

**Summary:**

This paper studied jailbreaking prompts generated by text optimizers that bypass alignment, discussed the effectiveness of threat models, performance of baseline defense techniques, and the differences between LLM security and computer vision. This paper studied detection based on perplexity, input preprocessing through paraphrasing and retokenization, and adversarial training. The effectiveness and limitations of each baseline defense were studied through extensive experiments. This paper pointed out that the weakness of existing text optimizers and the high cost of optimization make standard adaptive attacks more challenging for LLMs. The paper discussed the future research directions.

**Strengths:**

I think this work well analyzed three defense methods against jailbreak attacks that bypass moderation and alignment. This paper presented numerous experimental results to support these points and further discussed both the potential for attack methods to bypass the defense methods and the effectiveness of defense methods in detecting attacks.

**Weaknesses:**

The threat model is limited to only one type. It's advisable to incorporate additional models to validate the effectiveness of the three defense methods.

The paper did not propose a comprehensive method to address the shortcomings of the three defense methods. While the paper suggested that combining preprocessing defenses with detection defenses might be better , no specific approach or experimental result was provided.

Mixing existing datasets for adversarial training did not provide a rigorous defense across the entire search space.

**Questions:**

Please address weaknesses above.

---

> ### Author Response · Authors · 2023-11-16
> **Response to Reviewer Ywox**
>
> Thank you for your time, Reviewer Ywox!
>
> > threat model is limited
>
> We have now added an additional non-universal threat model, where the attacker has again access to the model weights, but now optimizes an attack for a single target string. As before, the attack does not have access to the system, so the defender can put preprocessing and post processing defenses in the pipeline. With this threat model, we optimize one string on the attacked model (Vicuna-v1.1), exploring the adaptive attack.
>
> | PPL Weight | Non-Attack ASR | ASR w/o filter | Passed PPL | Passed WinPPL | Total ASR (PPL Filter) | Total ASR (WinPPL Filter) |
> | ---------- | -------------- | -------------- | ---------- | ------------- | ---------------------- | ------------------------- |
> | 0          | 5%             | 100%           | 0%         | 0%            | 0%                     | 0%                        |
> | 0.1        | 5%             | 56%            | 50%        | 12%           | 14%                    | 0%                        |
> | 0.2        | 5%             | 12%            | 94%        | 56%           | 12%                    | 2%                        |
>
> > Comprehensive method
>
> We believe that each defense individually tells us something about the attack, and thus, should be studied individually to have a clearer scientific understanding of their merits. Nevertheless, in practice we agree that using a combination of defenses would be most practical. For example, the windowed perplexity filter and the paraphrase defense, i.e. paraphrasing only prompts that do not pass the perplexity filter.
>
> > Mixing existing datasets for adversarial training
>
> We agree. We believe there is still a lot of research to be done in this space. However, we do believe that we cannot "copy and paste" from how we did things in the CV domain. To round out our evaluation, we have now added an additional experiment where we mix in harmful prompts with random pre-crafted (already optimized) adversarial suffixes to instruction tuning dataset. More specifically, instead of generating an adversarial suffix for each example (as this is prohibitively expensive), we use precomputed suffixes optimized using a variety of different prompts. This is another approximation of the classical adversarial training. We see that this type of training does reduce the attack success rate (ASR) for attacked prompts bringing the performance of the attack down from 96% to 92%. However, we see that the unattacked malicious prompts did change with 95% of malicious prompts outputting bad behavior.
>
>
> | Model                         | ASR |
> | ----------------------------- | --- |
> | Alpaca Baseline               | 96% |
> | Alpaca + FT w/ adv. suffixes  | 92% |
>
> Please let us know if you there are any follow up questions.

---

> > ### Comment · Reviewer_hCXb · 2023-11-21
> >
> > I thank the reviewer for the response. Most of my concerns are addressed, and I am inclined toward increasing my rating score. I'll finalize my rating after discussing it with other reviewers and area chair in the later phase.

---

### Official Review · Reviewer_hCXb · 2023-10-31

**Soundness:** 3 good
**Presentation:** 3 good
**Contribution:** 3 good
**Rating:** 5
**Confidence:** 4

**Summary:**

Extending the studies in standard adversarial machine learning domains, this paper explores the effectiveness of three different mainstream defense methods against jailbreak attacks for LLM, including perplexity detection, input reprocessing via paraphrasing/retokenization, and adversarial training (more precisely, data augmentation with red-teaming prompts, instead of minmax-based training). Detailed experiments and discussions on the robustness-utility tradeoffs for each case are provided.

**Strengths:**

1. This is a good and timely study to help researchers understand how many lessons learned form computer vision domains for adversarial robustness can be transferred to LLMs.
2. The paper is easy to follow and provides abundant empirical results

**Weaknesses:**

The major concern that prevents me from giving a straight recommendation for acceptance is that the evaluation is based on the GCG attack proposed (Zou et al., 2023), which is a universal attack that learns to append the same adversarial suffix to every prompt for jailbreak attempts. However, from the computer vision domain, it is also known that universal attacks are easier to detect/defense than individually optimized adversarial examples (Finding a universal adversarial perturbation is mathematically more difficult than finding a specific adversarial perturbation for one image). Therefore, the conclusion made by this paper may be different if one considers the GCG attack on every prompt separately, instead of learning to do a universal attack.

Here are my suggestions:
1. I hope the authors can come up with new experiments that evaluate individual GCG attacks, and check if the conclusion holds or not. If also seems to me that the attacker may break the perplexity detection in the adaptive attack setting if the goal is to find a low-perplexity adversarial suffix for only one prompt, instead of many prompts simultaneously.
2. On adaptive attack against paraphrasing, can the authors check if appending the same adversarial suffix to the paraphrased prompt can regain the attack effectiveness or not?
3. On Sec. 4.3 - I hesitate to agree the studied method is "robust optimization". While it is true that the terminology of adversarial training (via augmenting with some (non-optimized) adversarial examples in the training loss) is consistent with what is proposed by Goodfellow et al. in 2013/2014, the evaluated scheme is different from the more popular minimax-based adversarial training method proposed by Madry et al., following the practice of robust optimization. It also seems that the studied method only augments with the human-crafted adversarial prompts to fine-tune an LLM, instead of iteratively generating new adversarial prompts during training. Therefore, I suggest removing the use of "robust optimization" in that paragraph, and making the message clear that this observation does not imply the result of minimax-based adversarial training method (which is unclear what is the best way to define and execute, as the authors pointed out). Otherwise, this section may give the incorrect message that "robust optimization" does not give strong robustness of LLMs against jailbreak attacks.
4. [Minor] There are some recent papers (after the ICLR submission deadline) that propose improved (automated) jailbreak prompt generations with high influence to bypass perplexity-based detections, so perplexity-based detections may not be as strong as the paper claimed (aka the defense can be already broken). However, I understand those results are concurrent to this study, and I won't take this point into my final rating.

**Questions:**

1. Will the same conclusion hold if one considers non-universal GCG attacks? That is, run GCG attack separately for each tested jailbreak prompt.
2. If appending the same adversarial suffix to the paraphrased prompt, can the attack remain effective?

---

> ### Author Response · Authors · 2023-11-16
> **Response to Reviewer hCXb**
>
> Thank you for your time, hCXb!
>
> > non-universal GCG attacks
>
> We have now added a non-universal threat model, where the attacker has again access to the model weights, but now optimizes the attack for a single string. As before, the attacker does not have access to the system, so the defender can put preprocessing and post-processing defenses in the pipeline. With this threat model, we optimize a single string on the attacked model (Vicuna-v1.1), exploring the adaptive attack.
>
> | PPL Weight | Non-Attack ASR | ASR w/o filter | Passed PPL | Passed WinPPL | Total ASR (PPL Filter) | Total ASR (WinPPL Filter) |
> | ---------- | -------------- | -------------- | ---------- | ------------- | ---------------------- | ------------------------- |
> | 0          | 5%             | 100%           | 0%         | 0%            | 0%                     | 0%                        |
> | 0.1        | 5%             | 56%            | 50%        | 12%           | 14%                    | 0%                        |
> | 0.2        | 5%             | 12%            | 94%        | 56%           | 12%                    | 2%                        |
>
> However, we find that these attacks, while stronger, still fail to break the perplexity filter, leading to a total ASR of 2% in with the optimal weight on the perplexity parameter.
>
>
> > same adversarial suffix to the paraphrased prompt
>
> This unfortunately would not be possible for the attacker in a realistic threat model. Since the attacker can only send a query to the system and the paraphrase is done internally over the query, the attacker does not have access to the prompt after the paraphrase.
>
> > Recent Attacks
>
> It is worth mentioning that these attacks did not consider the window perplexity nor paraphrase defense, which from our experiments we have found to be much stronger.
>
> Please let us know if there are any follow-up questions.

---

### Official Review · Reviewer_9hm2 · 2023-10-31

**Soundness:** 2 fair
**Presentation:** 2 fair
**Contribution:** 2 fair
**Rating:** 3
**Confidence:** 4

**Summary:**

The paper discusses simple baseline defenses against a single gradient-based adversarial attacks against Large Language Models (LLMs). These include perplexity based filtering, paraphrasing via another LLM, and an attempt at adversarial training.

**Strengths:**

Some of the results (in particular, regarding perplexity filtering) are interesting.

**Weaknesses:**

I find the motivation, ideas, and conclusions derived in this paper to be problematic for many reasons. I will also provide individual issues for each of the defenses studied, below.

- (Overview) **Lack of Proper Motivation and Significance**: The paper raises many questions (such as in the abstract: *"What threat models are practically useful in this domain?"* and *"How do baseline defense techniques perform in this new domain?"*) but fails to provide concrete answers for these to the reader. The paper ends on more unanswered questions (Section 5.1). This is due to a lack of proper motivation: the paper from the start is set up to only study the proposed baseline defenses which are either incompletely or trivially designed (I will discuss this later). Although the authors attempt to draw general conclusions, since there is only one attack proposed by Zou et al [1] for LLMs, the results cannot be interpreted generally (I discuss this next). Finally, the evaluation setups are inconsistent and keep varying between experiments, and smaller LLMs are mostly considered (discussed below). In sum, the paper feels incomplete and in my opinion does not provide substantial evidence for the initial claims made by the authors which is why I am leaning towards rejection.

- **Single-Attack Study Cannot Generalize**: The paper draws many *general* conclusions from the derived results, such as indicating the strengths or weaknesses of a particular baseline defense (e.g. end of Section 4.1: *"However, perplexity filtering is potentially valuable in a system where high perplexity prompts are not discarded, but rather treated with other defenses, or as part of a larger moderation campaign to identify malicious users"*). However, this is an untenable claim, as all the baseline defense experiments are conducted on a single optimization attack proposed by Zou et al [1] and we do not know the space of adversarial LLM samples that exist to a clear extent. Clearly, the scope of the work is fairly limited given this fact. Owing to these reasons, the paper seems incomplete and almost *too early* as there should be more attacks proposed first by the community to derive a general trend for baseline defenses. As an analogue, Carlini et al's [2] seminal paper on guidelines for defenses in deep neural networks was written after a number of attacks had been proposed.

- **Inconsistent Experimental Setups**: The paper has multiple experiments, but all of these are conducted on a different set of LLMs. For Section 4.1 the authors use Falcon-7B, Vicuna-7B, Guanaco-7B, ChatGLM-6B, and MPT-7B. For experiments in Section 4.2, Alpaca-7B is also introduced; why wasn't this used in 4.1 experiments? For Section 4.3 Llama1-7B and Alpaca are considered, and the other models are dropped. This inconsistency in experimental setup and evaluation is a major issue and hinders readers from getting a clear picture of the results. This issue also extends to other evaluation aspect of the work, such as in Section 4.1, nowhere do the authors mention what the actual value of the threshold $T$ for the perplexity filter is for most of the experiments. They just mention that it is the upper bound of perplexity on the AdvBench dataset which makes it hard to contextualize other results with respect to those of Figure 2 (when the threshold is varied).

- **Evaluation Only on Weaker Models**: Even though the original attack approach of Zou et al [1] was evaluated on multiple black-box LLMs such as GPT-3.5-Turbo, PaLM-2, and GPT-4, the authors never consider these powerful models in experiments. In fact, most of the experiments are only localized to smaller LLMs roughly around the 7B parameter size. If the issue is API or black-box access, in my view, the defense approaches can still be applied as a preprocessing step. Even then, as Llama-2-13B and Llama-2-70B models are available open-source, this issue can actually be bypassed. A lack of evaluation on powerful and more relevant models is also a major drawback of the work.

- **Issues with Each Defense Baseline**:
    - **Perplexity Filter**: Regarding the perplexity filter, the description by the authors of its efficacy (last paragraph Section 4.1: *"the defense succeeds.."* and others) is in contrast with the results obtained. As Figure 1 shows, the Attack Success Rate (ASR) is greater than 50% for the Guanaco and Falcon LLMs even as $\alpha_{ppl}$ is increased. Also, ChatGLM and MPT LLMs are the only ones for which ASR tends to 0, but their performance initially is close to 0 to begin with. More importantly, even after the filter, if 20% of the prompts bypass the windowed perplexity filter, why can the attacker not just reproduce prompts similar to these (e.g. via paraphrasing) and attack the LLM with a large volume of such prompts? It also excessively targets benign prompts and I am not convinced that there is a good approach for choosing the threshold $T$. Due to these reasons, I believe this baseline is too trivial and does not convery any useful information about efficacy and complexity of optimizer attacks for LLMs.

    - **Paraphrasing Prompts**: The issue with this defense is that if the defender can utilize a more powerful LLM for paraphrasing (such as GPT-3.5-Turbo as the authors have done), why can they not directly use that model for their LLM related tasks? It would make more sense to directly use ChatGPT instead of filtering responses using ChatGPT and then using Vicuna. What would be interesting and more viable, would be to use a more traditional paraphrasing approach for the adversarial prompt and then assessing whether ASR can be lowered. In its current form, the defense does not seem tenable as a baseline.


    - **Attempting Adversarial Training**: The approach employed by the authors in Section 4.3 is an approximate attempt at adversarial training. It is evident why it doesn't work well, as the inner adversarial optimization step can only use human generated adversarial prompts instead of optimizer generated ones, due to computational issues. In this manner, this approach is incomplete and already unsatisfactory as a baseline. In my opinion, it cannot even constitute a negative result because the approach is not sufficiently close to true adversarial training. This also relates to my earlier point on the work being *premature*. In this case, it would actually be beneficial to improve the optimization approach used for attack samples to enable adversarial training of LLMs.



___
___
___
References:
1. Zou, Andy, et al. "Universal and transferable adversarial attacks on aligned language models." arXiv preprint arXiv:2307.15043 (2023).
2. Carlini, Nicholas, et al. "On evaluating adversarial robustness." arXiv preprint arXiv:1902.06705 (2019).

**Questions:**

- Is there a reason the authors did not study more powerful models such as GPT-3.5-Turbo, BARD (PaLM-2), Llama-2 and GPT-4? The biggest strength/finding of the paper by Zou et al [1] is that the attack transfers (in a black-box fashion) to these other models.
- Please feel free to reply to any of the weaknesses listed in the previous section.

___
___
___
References:
1. Zou, Andy, et al. "Universal and transferable adversarial attacks on aligned language models." arXiv preprint arXiv:2307.15043 (2023).

---

> ### Author Response · Authors · 2023-11-16
> **Response to Reviewer 9hm2 (part 1/2)**
>
> Thank you for your time, 9hm2!
>
> >  Single-Attack Study Cannot Generalize -- conclusions limited from studying one attack threat model
>
> We do measure defense efficacy on only one attack, we have modified the claims in the first paragraph of Section 4 to be more clear to reflect your point about the timing and maturity of the field. We feel that it is a good time to start considering defenses and furthermore that the Zou et al. attack we consider is a representative example of existing methods. There are several existing methods (Wallace et al. 2019--https://arxiv.org/abs/1908.07125, Jones et al. 2022--https://arxiv.org/abs/2303.04381), but it is clear that Zou et al. is the strongest, so for benchmarking defenses we choose to use the best offense available. Zou et al. show that they strengthen the optimization process and jailbreak powerful modern LLMs. Notice that all of these optimization-based jailbreaks include a suffix/prefix that makes them similar from the detection standpoint.
>
> Also, we have added some experiments with stronger, non-universal threat model. We find that defenses through perplexity filtering still hold up. As before, in these experiments, we explore a threat model where the attacker has access to the model weights. But here, the attacker optimizes an attack against a single target string. However, the attacker does not have access to the system, so the defender can put preprocessing and post-processing defenses in the pipeline. With this threat model, we optimize one string on the attacked model (Vicuna-v1.1), exploring this adaptive attack. We see that even in this case it is very difficult to completely break the windowed perplexity defense without degrading the potency of the attack.
>
> | PPL Weight | Non-Attack ASR | ASR w/o filter | Passed PPL | Passed WinPPL | Total ASR (PPL Filter) | Total ASR (WinPPL Filter) |
> | ---------- | -------------- | -------------- | ---------- | ------------- | ---------------------- | ------------------------- |
> | 0          | 5%             | 100%           | 0%         | 0%            | 0%                     | 0%                        |
> | 0.1        | 5%             | 56%            | 50%        | 12%           | 14%                    | 0%                        |
> | 0.2        | 5%             | 12%            | 94%        | 56%           | 12%                    | 2%                        |
>
> > Inconsistent Experimental Setups
>
> Due to compute limits, instead of choosing a 13B model and a 7B model, we were forced to choose two 7B parameter models, weakening the attack. Thus, the models chosen in each section were chosen by if they started with an initially high ASR to begin with, studying models such as MPT-Chat or ChatGLM that have low ASR, even before any defense, would tell little about the defense as there are 100 test examples. In the case of perplexity, we can individually examine whether the attack bypassed the model or not regardless of the ASR. For Alpaca, the PPL and WinPPL detect the original attack 100% of the time.
>
> For Section 4.3, we had to finetune a model. Thus, we chose the most popular finetuning dataset--Alpaca, which was originally finetuned on LLaMA-1.
>
> >Evaluation Only on Weaker Models:
>
> Our choice of which models to use in evaluating defenses has three justifications. (A) we use a similar set to Zou et al. (B) Strong models that are naturally impervious to these attacks provide little room to demonstrate the efficacy of a defense. (C\) Closed models that live behind paywalls and may change in time are hard to do reproducible scientific experimentation with. Since the release of Zou et al. some of the black-box chatbot systems have added additional defenses (at least it seems so, given that attacks proposed in Zou et al. do not work as well anymore). Of course, we don't know if these are pre or post-processing (or any other type of defense) which leads to the two issues mentioned above: attacks show near-zero ASR, and reproducible experiments are impossible.
>
> > Issues with Each Defense Baseline -- Perplexity Filter
>
> We say the defense succeeds in its goal to limit the effect of the attacked prompt. Now, the model might be badly aligned to begin with, which is the case for Guanaco and Falcon making the ASR close to the unattacked version of the malicious prompts. In the reverse case, MPT-Chat and ChatGLM have good alignment making them hard to attack to begin with.
>
> For choosing a good value of $T$, we plotted the Pareto frontier curve to show how a user might choose a tolerance value. Below are the thresholds used for Figure 6 and Figure 7 in the Appendix and Table 2 in the main paper.
>
> |  |  Vicuna-v1.1 | Guanaco | Falcon-7B-Instruct | ChatGLM-6B     |  MPT-7B-Chat     |
> | ----------- | ------- | -------------- | --- | --- | --- |
> | PPL Threshold        | 4.354    | 4.665           |  5.292   |  7.988   |   5.074  |
> | Window PPL Threshold        | 6.651    | 6.403           |  7.996   |   12.375  |  7.792   |

---

> > ### Author Response · Authors · 2023-11-16
> > **Response to Reviewer 9hm2 (part 2/2)**
> >
> > > Issues with Each Defense Baseline -- Paraphrasing Prompts
> >
> > ChatGPT is not the most powerful paraphraser available. There are more powerful ones like Dipper. But you make a good point and to clarify the influence of the external paraphrasing model we have now run additional experiments with Vicuna-v1.1. In these experiments, we use the same model to both paraphrase and generate responses. We see in this case that the ASR of the jailbreak drops from 80% to 13%, compared to 5% for ChatGPT. Looking at the qualitative examples we see that in many cases the adversarial prompt is removed similar to ChatGPT.
> >
> > It is interesting to discuss practical use cases though, and we can imagine having a particular model on which they need to do inference (maybe because of domain expertise), and paraphrasing with a different (maybe stronger in the general sense). In such a scenario, having the paraphrasing as a separate model would be reasonable.
> >
> > > Issues with Each Defense Baseline -- Attempting Adversarial Training
> >
> > This section was to help illustrate that traditional techniques from CV cannot be directly applied to the language models. Thus, we tried some approximation of robust optimization and we agree that our version does not work (we identify in the paper that it is a weak defense).
> >
> > A point that we wanted to raise, but maybe did not clarify enough is that these approximations are necessary. In vision, adversarial attacks can succeed within 2-7 PGD steps, making adversarial training feasible. Yet for language, there is no success using the attack of Zou et al. for less than several thousand model evaluations.
> >
> > To round out our approximation strategies, we have now added an additional experiment where we mix in harmful prompts with random pre-crafted (already optimized) adversarial suffixes to the instruction tuning dataset. More specifically, instead of generating an adversarial suffix for each example (as this is prohibitively expensive), we use precomputed suffixes optimized using a variety of different prompts. This is another approximation of the classical adversarial training. We see that this type of training does reduce the attack success rate (ASR) for attacked prompts bringing the performance of the attack down from 96% to 92%. However, we see that the unattacked malicious prompts did change with 95% of malicious prompts outputting bad behavior.
> >
> > | Model                         | ASR |
> > | ----------------------------- | --- |
> > | Alpaca Baseline               | 96% |
> > | Alpaca + FT w/ adv. suffixes  | 92% |
> >
> > Please let us know if there are any follow-up questions.

---

> > > ### Comment · Reviewer_9hm2 · 2023-11-20
> > > **Response to Rebuttal**
> > >
> > > I would like to express my gratitude to the authors for their rebuttal. However, after going through the response, my original concerns still largely remain. I provide some more details below:
> > >
> > > * **Study Generalization Issues**: I find that despite the authors stating that Zou et al might be the strongest attack, my concern still holds. The work undertaken in the paper cannot generalize as long as a single attack is considered, and I think it would be hard to find a lot of other work in the adversarial attack or defense field that only ever considers a single attack. Moreover, new attack approaches are being proposed at the moment (for e.g. https://arxiv.org/abs/2310.13345), which brings me to the second issue of the work being somewhat premature.
> > > * **Experimental Issues**: While I understand that the authors might have faced compute issues, since the work is aiming to generalize across LLMs, the lack of comparison on 13B and larger models should be studied. Similarly, if the authors think stronger models (or ones behind paywalls) are not susceptible to the Zou et al attack, then they can themselves be used as in-processing defense approaches against the attack. I think this dilutes the motivation for the current work quite significantly.
> > > * **Issues with Individual Defenses**: Thank you for providing details on the perplexity filter baseline and additional results for the paraphrasing baseline. However, I am not convinced of the adversarial retraining strategy. Philosophically, despite the work aiming to propose an analogue to adversarial training for LLMs, it fails to do so, which can neither be seen as a negative or a positive result. For e.g. it is not clear to me that it is not possible to design better or improved approaches for adversarial LLM training, it is just that the authors do not have a good solution for it at the moment (i.e. lack of improved optimization strategies). Since the authors raise and motivate this defense themselves, not following through (in my opinion) is not good enough. I am also not sure of trying to connect the work to vision based attacks, as LLMs are very different conceptually, and hence, it shouldn't be the case that the attacks would work similarly. They are neither vision models nor are they similar in task output or predictive capability.
> > >
> > > Given the points above, I would like to maintain my current score.

---

> ### Author Response · Authors · 2023-11-21
> **Reply to Reviewer 9hm2**
>
> Thank you for your response.
>
> > Study Generalization Issues
>
> The threat models we explore are optimized attacks against aligned language models. This particular type of attack is against the alignment of the model and is different from GLUE attacks that were explored in https://arxiv.org/abs/2310.13345. Additionally, it is not uncommon to study a single popular attack in defense papers, e.g. https://arxiv.org/abs/1312.6199, which is well-regarded by the community.
>
> In this paper, we studied a number of defenses in our work to understand (1) what are the takeaways from Computer Vision (CV) and what types of defenses are stronger in LLMs than CV (2) how should we be thinking of designing future attacks, i.e under what lens taking the lessons learned from CV. Additionally, we would argue that it is never too early to consider defenses as they help set the bounds of future attacks.
>
> > Experimental Issues
>
> We do know from Zou et al. that Vicuna-7B and Vicuna-13B have the same ASR. Thus, larger models are not more robust to these attacks when trained with the same procedure, and therefore, studying Vicuna-7B is still very insightful. Furthermore, we want to reiterate that black-box API models are not meaningful objects of study, as these systems may already include defensive mechanisms, which will obstruct a clear scientific evaluation.
>
> >Issues with Individual Defenses -- Adv Training:
>
> We are glad we clarified questions that arose for the ppl defense and paraphrase defense. From the response above, it seems that there are still some questions about adversarial training. The study of adv training is inherited from the many papers that study adversarial attacks in CV, and because of this, we wanted to discuss and outline the challenges of this type of defense for aligned language models. We provide a baseline defense for adv training that we find not incredibly effective at this moment. As training with adv examples is a very natural avenue of approaching the problem, we wanted to include this section as it is a natural baseline for defending against attacks and showcases an important line of research.
>
> We appreciate the continued engagement, Reviewer 9hm2.

---

> > ### Comment · Reviewer_9hm2 · 2023-11-23
> >
> > Thank you for your additional responses. I provide more details below.
> >
> > > This particular type of attack is against the alignment of the model and is different from GLUE attacks that were explored in https://arxiv.org/abs/2310.13345
> >
> > Yes, I am aware, I was referring to the fact that new attacks against LLMs are still being proposed at the moment. By the same logic attacks against the current threat model should also be proposed.
> >
> > > Additionally, it is not uncommon to study a single popular attack in defense papers, e.g. https://arxiv.org/abs/1312.6199, which is well-regarded by the community.
> >
> > The paper by Szegedy et al is a seminal work of the field, and arguably one of the first works studying adversarial attacks in deep neural networks. More than half of the paper also studies neural network properties and is not concerned with attacks. The analogue here is if the authors had proposed a single field-defining attack that improved upon the work by Zou et al I would have considered it a valid contribution. However, the paper is studying baseline defenses against a _single_ attack. I do not think this comparison holds. Irrespective of all this, my concerns regarding generalization have not been alleviated.
> >
> > > what are the takeaways from Computer Vision (CV) and what types of defenses are stronger in LLMs than CV
> >
> > This has also not been satisfactorily explained. I am unsure of why LLM attacks are being related to computer vision, the models are completely different. Attacks against NLP models have long since utilized adversarial training and other approaches in attack optimization [1] and since clearly LLMs are closer to the text (instead of vision) modality, it does not make sense to compare attacks/behaviors.
> >
> > > We do know from Zou et al. that Vicuna-7B and Vicuna-13B have the same ASR. Thus, larger models are not more robust to these attacks when trained with the same procedure, and therefore, studying Vicuna-7B is still very insightful.
> >
> > Well this is again a major assumption. Just because the performance of Vicuna across two model classes is similar does not imply the statement made above (larger models are not more robust to these attacks) holds across LLMs and for different sizes. It would have been nice to see evidence by way of actual experiments conducted across larger models. If the authors then obtain the same results, I would be inclined to agree with the statement above. Right now based on current evidence provided, I am not sure, and leaning towards the contrary.
> >
> > > We provide a baseline defense for adv training that we find not incredibly effective at this moment.
> >
> > My original point still stands. It is not that adversarial training is not possible for LLMs, it is just that the current approaches proposed in this work have not focused on finding ways of solving the problem. Adversarial training might be fairly easy for LLMs if some better optimization approach was proposed for this purpose, but we do not know that. In that sense, I am not convinced.
> >
> >
> > ___
> > 1. Morris, John, et al. "TextAttack: A Framework for Adversarial Attacks, Data Augmentation, and Adversarial Training in NLP." Proceedings of the 2020 Conference on Empirical Methods in Natural Language Processing: System Demonstrations. 2020.

---

### Meta-Review · Area_Chair_jGAc · 2023-12-03

**Metareview:**

This paper assesses the extent to which three common defense methods against adversarial attacks apply to large language models, attempting to derive general conclusions on their robustness-utility trade-off.

It is timely and important to systematize research on adversarial attacks against LLMs. However, the paper might not be mature enough to be published, as general conclusions are drawn from a limited number of attacks, the defenses might not be strong enough, and the conclusions are for the time being too general to be of practical use. Moreover, while I agree with the stance that scientific work should not necessarily engage with paywall-protected, poorly documented models, it is disappointing that the authors could not experiment with models larger than 7B.

**Justification For Why Not Higher Score:**

While the paper proposes a promising direction, the current setup is too limited to draw general, useful conclusions.

**Justification For Why Not Lower Score:**

N/A

---

### Decision · Program_Chairs · 2024-01-16

Reject